# Emergence and Spread of *Piscine orthoreovirus* Genotype 3

**DOI:** 10.3390/pathogens9100823

**Published:** 2020-10-07

**Authors:** Juliane Sørensen, Niccolò Vendramin, Camilla Priess, Dhamotharan Kannimuthu, Niels Henrik Henriksen, Tine Moesgaard Iburg, Niels Jørgen Olesen, Argelia Cuenca

**Affiliations:** 1Unit for Fish and Shellfish Diseases, National Institute of Aquatic Resources, Technical University of Denmark, 2800 Kgs. Lyngby, Denmark; jusor@aqua.dtu.dk (J.S.); niven@aqua.dtu.dk (N.V.); capr@aqua.dtu.dk (C.P.); timi@aqua.dtu.dk (T.M.I.); njol@aqua.dtu.dk (N.J.O.); 2NMBU, Department of Food Safety and Infection Biology, Faculty of Veterinary Medicine, Norwegian University of Life Sciences, 0454 Oslo, Norway; dhamubfsc@gmail.com; 3Dansk Akvakultur, 8200 Aarhus, Denmark; niels@danskakvakultur.dk

**Keywords:** *Piscine orthoreovirus* genotype 3, double-stranded RNA virus, epidemiology, rainbow trout, surveillance program, phylogenetic analysis, full genomes

## Abstract

*Piscine orthoreovirus* (PRV) is a relevant pathogen for salmonid aquaculture worldwide. In 2015, a new genotype of PRV (genotype 3, PRV-3) was discovered in Norway, and in 2017 PRV-3 was detected for first time in Denmark in association with complex disease cases in rainbow trout in recirculating aquaculture systems (RAS). To explore the epidemiology of PRV-3 in Denmark, a surveillance study was conducted in 2017 to 2019. Fifty-three farms, including both flow through and RAS, were screened for PRV-3. Of the farms examined, PRV-3 was detected in thirty-eight (71.7%), with the highest prevalence in grow-out farms. Notably, in Denmark disease outbreaks were only observed in RAS. Additionally, wild Atlantic salmon and brown trout populations were included in the screening, and PRV-3 was not detected in the three years where samples were obtained (2016, 2018, and 2019). Historical samples in the form of archived material at the Danish National Reference Laboratory for Fish Diseases were also tested for the presence of PRV-3, allowing us to establish that the virus has been present in Denmark at least since 1995. Sequence analyses of segment S1 and M2, as well as full genome analyses of selected isolates, did not reveal clear association between genetic makeup in these two segments and virulence in the form of disease outbreaks in the field.

## 1. Introduction

*Piscine orthoreovirus* (PRV) is a segmented double stranded RNA virus (dsRNA) belonging to the family *Reoviridae*, genus *Orthoreovirus*. PRV virions are non-enveloped particles with a double protein capsid with icosahedral symmetry [1]. The viral genome consists of ten RNA segments denoted S1-4 (S for small, ranging from 1.052 to 1.348 Kb), M1-3 (M for medium, ranging from 2.190 to 2.450 Kb), and L1-3 (L for large, ranging from 3.970 to 4.014 Kb) [1,2,3]. Currently, PRV cannot be cultivated in vitro, and only ex vivo infection of red blood cells allows certain replication [4]. 

To describe the different levels of variation within PRV throughout this paper, the terminology genotype is used for PRV-1, -2, and -3, and subtype to describe the groups within genotypes (e.g., PRV-3a and -3b) [5].

To date, three genotypes of PRV have been characterized: PRV-1, which has been shown to cause heart and skeletal muscle inflammation (HSMI) in farmed Atlantic salmon (*Salmo salar*) in Norway [6]; and has been detected in melanised foci in the fillet of farmed Atlantic salmon in Norway [7]. Furthermore, PRV-1 has also been associated with jaundice syndrome in Chinook salmon (*Oncorhynchus tshawytscha*) [8], but causality has not been confirmed by experimental studies [9,10]. PRV-2, which has been found in coho salmon (*Oncorhynchus kisutch*) in Japan where it causes erythrocytic inclusion body syndrome (EIBS) [11,12].

PRV-3 was discovered in 2013 during the diagnostic investigation of an unexplained event of increased mortality in a flow through farm producing rainbow trout (*Oncorhynchus mykiss*) in fresh water in Norway [13]. Later on, a similar disease outbreak was reported in brown trout in France [14]. In 2017, PRV-3 was detected in Denmark where it was associated with the onset of severe disease with high mortalities in farmed rainbow trout [3,15]. Furthermore, PRV-3 has been detected in association to disease outbreaks in Germany [16,17] and Scotland [3], and in samples from asymptomatic brown trout (*Salmo trutta*) in Italy, and in brown trout in Czech republic [3,18]. Since the findings of PRV-3 in Denmark in 2017, a number of studies aimed at elucidating the role of PRV-3 in disease have been carried out. Initial experimental work showed efficient horizontal transmission of PRV-3 and development of heart pathology resembling HSMI in rainbow trout after viral peak in heart tissue [19]. The same study showed a certain host preference for rainbow trout compared to Atlantic salmon. Furthermore, development of heart pathology resembling HSMI was observed after viral peak in target organs in an experimental infection with purified viral particles in rainbow trout [20] further corroborating the role of PRV-3 as causative agent of heart pathology in rainbow trout. Additionally, PRV-3 has been associated with proliferative darkening syndrome (PDS) [17], a condition affecting wild brown trout in the pre-Alpine rivers of Austria, southern Germany, and Switzerland with nearly 100% mortality [17]. However, the causative relationship between PRV-3 and PDS remains unclear [21].

Overall, *Piscine orthoreoviruses* are widespread in their respective host niches, and therefore viral genome detection by PCR does not necessarily imply a disease outbreak. PRV-1 is present in nearly every cohort study of farmed Atlantic salmon in the marine phase [6], and PRV-3 has been detected in non-diseased adult rainbow trout in Norway [22], and in several European countries from both disease outbreaks and asymptomatic fish [3]. Although with only a few PRV-3 sequences available so far, a preliminary phylogenetic analyses of PRV-3 S1 segment has clustered the original Norwegian isolate separately from the other PRV-3 sequence isolates from Europe and Chile [3].

The first aim of this study is to assess the presence of PRV-3 in Denmark, and to highlight risk factors involved in disease onset in fish farms. A voluntary surveillance program was conducted, where the presence of PRV-3 was assessed in Danish farms for a one and a half year period. The program included both rainbow trout and brown trout farms, and farms in all different stages of production from hatchery to grow-out phase. The farms also included a whole range of water recirculation systems, from open flow through to intensive RAS, covering all the different kinds of aquaculture production in Denmark. In order to backtrack the presence of the virus in the country, we also screened archived material of the Danish National Reference Laboratory for Fish Diseases back to 1995. The full genome was sequenced for a small number of isolate representatives, both for recent and historical samples. 

In an attempt to trace the spatio-temporal dynamics of PRV-3 in Denmark, sequences from the S1 and M2 segment of PRV-3 positive surveillance cases were obtained. These two segments were chosen because it has been suggested that virulence in PRV-1 is linked to these segments [23]. Indeed, phylogenetic analyses of PRV-1 sequence isolates found possible co-evolution of segments S1 and M2, grouping all sequence isolates causing HSMI, with exception of PRV-1a which is associated with HSMI-like pathology [24], in these two segments only. It is hypothesised that mutational differences in these two segments might account for differences in virulence, but not pathogenicity, between subtypes of PRV-1 [23]. Thus, the second aim of this study is to test whether this is also true for PRV-3.

## 2. Results

### 2.1. Surveillance Program for the Presence of PRV-3 in Denmark

A voluntary surveillance program targeting PRV-3 was conducted in collaboration with the Danish Aquaculture Organisation (Dansk Akvakultur, Aarhus, Denmark) and private veterinarians. The program started in fall 2017 and finished in early spring 2019; it included 53 farms (30% of the 175 active Danish farms producing salmonids in the country in 2018). During the time span of the program, 75 samplings were carried out, and 2031 samples were analyzed for the presence of PRV-3 by RT-qPCR. In order to accommodate the large number of samples, a testing scheme was implemented in which samples were tested in pools consisting of five fish. The pooling effect on assay sensitivity was evaluated prior to start of the surveillance program (Appendix A).

At each sampling, heart and spleen were collected from around 30 fish individually (the number of fish sampled varied in some samplings). In addition, information about farm characteristics were recorded: production type (broodstock or production), type of water recirculation (from flow through to intensive recirculating aquaculture systems), presence of biofilters, fish species produced, age and history of the fish, and tank number, along with an assessment of the health status of the fish. The amount of information provided per case regarding the fish age, size, history and tank number varied from case to case, however.

The program was organized in a way that the veterinarians could also submit samples from the same farm at different time points, as well as when increased mortality was recorded on the farm or when fish were showing clinical signs (i.e., abnormal swimming behaviour, increased gill movement etc.). After each sampling, results were made available to the fish farmers and the private veterinarian responsible for the sampling. Samples from wild salmonids (Atlantic salmon and sea trout) were obtained in 2016, 2018, and 2019 from Danish Centre for Wild Salmon (Danmarks Center for Vildlaks, DCV), of which all samples tested negative for PRV-3 (tested as individual samples).

### 2.2. Presence of PRV-3 in Association with Farm Characteristics

Out of the 53 farms included in this study, 38 (71.7%) were positive for PRV-3, although in three of them PRV-3 was present at very low levels. In these three cases, results were corroborated by testing fish individually, revealing that only one or two fish out of 30 were positive with high Ct values (≥35 Ct). Importantly, only five farms have been tested negative across multiple samplings.

As the surveillance program unveiled that PRV-3 has a high prevalence in Denmark, it became necessary to investigate if the spread of the virus was natural (e.g., following a particular water catchment) or not. Mapping all farms included in the surveillance program revealed that there is no association between the geographical distribution and of the presence of PRV-3. Furthermore, PRV-3 is widespread within the most of the country, not following any clear phylogeographic pattern, as is expected by virus movement by trade of fish from one farm to another. 

Water recirculation appear to increase the risk of a farm being positive for PRV-3, as 100% of RAS farms with biofilters tested positive (18 out of 18), compared to 50% of flow through (11 out of 22, Fisher exact test *p* = 0.0003) (Figure 1a). Additionally, grow-out facilities display a higher risk of testing positive, with 87% (27 out of 31) being positive for PRV-3 compared to 47% (9 out of 19) broodstock farms (*p* = 0.0038) (Figure 1b).

To expand the knowledge on host susceptibility range and epidemiology, the study included testing of farms producing brown trout for restocking. Four out of five screened farms producing brown trout were positive. Two farms housed both rainbow trout and brown trout in their facilities. Interestingly, in one of these farms, only samples from rainbow trout were positive, while all samples from brown trout were negative for PRV-3.

### 2.3. Presence of PRV-3 and its Association with Disease

While PRV-3 has been detected in 38 of the farms included in the surveillance program, disease associated with detection of the virus has only been observed in 10 farms (approximately 26% of the positive farms). All the farms affected were RAS to different degrees.

Disease has never been observed in association with PRV-3 in flow through farms in Denmark, or in any of the brown trout farms included in the surveillance. 

### 2.4. Historic Samples from Clinical Disease Outbreaks

As PRV-3 is more prevalent in Denmark than initially expected, investigation of archived material from 1995 was conducted. All the archived samples from 1995 (*n* = 29) representing fish tissue from 25 rainbow trout farms (covering broodstock, production, and sea cages) originate from diagnostic investigations where infectious pancreatic necrosis virus (IPNv) was isolated in association with disease outbreaks. Eleven samples (38%) of these 1995 samples tested positive for PRV-3. 

### 2.5. Sequencing and Phylogenetic Analysss

For each PRV-3 positive farms, only one pool with the highest viral load was sequenced per sampling. 

Partial sequences of segment S1, encoding the σ3 protein (outer capsid protein, zinc metalloprotein) [1], and M2, encoding the µL protein (outer capsid protein, involved in membrane penetration) [1], were obtained for 51 and 34 isolates respectively. Additional sequences were obtained from Genbank (ncbi.nlm.nih.gov/genbank/). Phylogenetic analyses of PRV have largely been based on the S1 segment and therefore a major representation of PRV-3 sequences was available for this segment, including a number of Chilean PRV-3 sequences isolated from rainbow trout and coho salmon in 2013 and 2014. As very few sequences of the segment M2 are available in public databases, the number of isolates in the M2 dataset is considerably smaller (71 PRV-3 isolates in the S1 dataset, and 37 in the M2 dataset).

Figure 2a,b show the maximum likelihood tree of the S1 and M2 segment, respectively. The panel of sequences included in the phylogenetic analyses represent: (1) PRV-3 positive surveillance samples collected in Denmark from late 2017 to early 2019, (2) PRV-3 positive samples from Norway from 2016, (3) archived Danish samples from 1995, and (4) any available S1 and M2 sequence in GenBank except for sequences too short (about 50% size) to be included (accession numbers can be found in Appendix A). In total, six farms were represented by two and three isolates as they were sampled across several occasions (Figure 2a).

Consistent with previously shown [3], phylogenetic analyses based on segment S1 recovered both clades PRV-3a and PRV-3b as monophyletic with 93% and 95% bootstrap support, respectively (marked with bold in Figure 2a). Sequences from Norwegian isolates from 2016, published here for the first time, are closely related with the NOR/060214 within the PRV-3a clade. With exception of four sequences, all isolates from the surveillance program in Denmark, including the historical samples from 1995, are part of the PRV-3b clade, together with isolates from Chile and Europe. This clade encompassed isolates from brown trout, rainbow trout, and coho salmon. The placement of two brown trout isolates (DK/18-6340-33 and DK/18-5020-3) and two rainbow trout isolates (DK/18-4929-33 and DK/18-6518-30) is problematic, as they do not seem to belong to either PRV-3a or PRV-3b clades (Figure 2a marked in bold). In fact, their position in the phylogeny changes depending on which method was used to construct the ML tree (Figure 2a versus Appendix A). From S1 and M2 trees (Figure 2a,b), it is clear that, at least in segments S1 and M2, there is not a direct association between sequence and disease, in the sense that no particular clade encompassed disease causing isolates only. This lack of correlation is evident as isolates causing disease are widely inter-dispersed along the phylogeny of PRV-3, and isolates recovered from farms suffering severe disease outbreaks are identical to isolates from farms with no disease detected. No association between PRV-3 genetic makeup and host species was detected either. 

Additionally, no evidence of recombination was found within segment S1. In segment M2, weak evidence of recombination was found for sequence NOR/060214 with RDP4 [25], but this could not be corroborated when using GARD [26].

The current distribution and clustering of PRV-3 isolates in the two segments is reflected also in the haplotype network analysis of (Figure 3). The S1 segment recovered 22 haplotypes (Figure 3, Appendix A). The main dominant haplotype was DK4, including 29 (59%) of the Danish isolates included in this study, and spanning from 1995 to 2018. The S1 haplotype network also seems to indicate a single introduction event of Chilean isolates, but caution is required given the few number of Chilean isolates available, covering a very short time-span. As shown in Figure 2a, the haplotype corresponding to brown trout isolates DK/18-5020-3 and DK/18-6340-33 (haplotype DK1), and the haplotype corresponding to sequence isolates from two vertically integrated companies DK/18-6518-30 and DK/18-4929-33 (haplotype DK12), do not belong to the PRV-3b clade, being quite different from other PRV-3 isolates (DK/18-6340-33 and DK/18-5020-3 are identical, and the percent nucleotide identity is 96.2% and 95.7–96.1% to PRV-3a and PRV-3b, respectively. DK/18-4929-33 has a percent nucleotide identity of 97.3% to PRV-3a, and 88.3–96.3% to PRV-3b. Finally, for DK/18-6518-30 the nucleotide identity is 97.2% and 88.3–96.2% to PRV-3a and PRV-3b, respectively), and may indicate a different introduction event of PRV-3 within the country, or at least that two different genetic lineages are coexisting within the country.

### 2.6. Full Genome Analyses of PRV-3

Complete full genomes were obtained for three Danish PRV-3 isolates. To do so, and due to the lack of an in vitro replication method, infection experiments were done to propagate all PRV-3 isolates subject to full genome sequencing in vivo.

Full genome sequencing was done using two different methods. For the two samples from 2017 and 2018, RNA was isolated from plasma, and RNA was sequenced using on the Illumina HiSeq 4000 system. In the case of the sample of 1995, RNA was extracted from serum, but here a sequencing independent single-primer amplification (SISPA) method was used previous to library construction [27] and sequencing in Illumina MiSeq.

For the contemporary samples isolate DK/PRV317 and DK/PRV315, 156,819,216 reads and 105,445,446 reads were obtained, respectively (Appendix A). Of those, only about 0.1% of the reads belonged to PRV-3. De novo assembly, done with only 5% of the total data (7,840,890 and 5,272,273 reads, respectively), resulted in 1517 contigs for DK/PRV315 (19 > 1000 bp, N50 = 6), and 6935 contigs for DK/PRV317 (43 > 1000 bp, N50 = 14).

For the historical sample of 1995, 1,280,914 reads were obtained of which 966,924 (0.75%) belonged to PRV-3. The de novo assembly resulted in 50 contigs (14 > 1000 bp, N50 = 5). The largest obtained contig had a length of 3982, corresponding to the L1 segment of PRV-3.

Full genome phylogenetic analyses of the three Danish PRV-3 isolates were carried out including the two previously published PRV-3 genomes from rainbow trout in Norway and from coho salmon in Chile (Figure 4, refer to Appendix A for accession numbers). This sampling spans 23 years of evolution of PRV-3 (1995-2018). One unpublished German isolate from 2008 in brown trout was also included, even though not all segments were fully sequenced (partial sequences available for all segments except L2, Appendix A). Additionally, a number of PRV-1 isolates were included spanning a similar time period (1988–2014), as well as the only PRV-2 isolate where the full genome is available so far (Appendix A).

Phylogenetic analyses performed on individual segments were relatively consistent (see Appendix A figure S4 for phylogenetic trees of all 10 segments of the PRV genome). With exception of segment S2, all segments recovered the Norwegian isolate NOR/060214 as sister of a well-supported clade (100% BS) including PRV-3 isolates from Denmark, Germany and Chile. This is consistent with the suggestion of calling the clade containing the Norwegian isolate, and the clade containing isolates from Europe and Chile for subtypes PRV-3a and PRV-3b, respectively. Phylogenetic analyses performed using the S2 segment does not recover this dichotomy between the Norwegian and the Danish/German/Chilean isolates, and it places the Norwegian isolate within the PRV-3b clade (Appendix A) (99% BS). Whether this is caused by segment assortment needs to be explored further. 

## 3. Discussion

The aim of this study was to assess the prevalence, distribution, and genetic variability of PRV-3 in Denmark in order to (1) gain knowledge of the epidemiology of the virus, and assess whether there was a correlation between prevalence and type of farming system, and (2) whether there was a correlation between isolate sequence and disease as it has been suggested for PRV-1. By testing 2031 samples representing 53 salmonid farms we demonstrated that PRV-3 is widespread in Denmark. PRV-3 was detected in fish farms encompassing flow through and a variety of different recirculating aquaculture systems. The prevalence measured was 71.7% of the farms tested, which represent 30.3% of the total amount of active salmonid farms in Denmark in 2018 (175 active farms). Notably, clinical disease outbreaks associated with the detection of PRV-3 were seen in only 26% of the PRV-3 positive farms (10 out of 38 positive farms). Three farms have, however, had disease associated with the detection of PRV-3 on multiple occasions.

The distribution of positive farms does not reflect a natural spread of the virus, as positive farms are located throughout most of the country. The most obvious explanation is related to the trade of live fish, which can be asymptomatic carriers (as is observed in fish testing positive for the virus without showing clinical disease). By molecular tracing, we could document that there has been known transport of fish from the farm associated with isolate DK/18-5650-58 to the farms associated with the isolates DK/18-3758-33, DK/18-4260-7, and DK/18-1766-19. These isolates are identical or very similar to each other, indicating that transport of contaminated and non-disinfected eggs or infected fish can equal transport of the virus.

Our data showed a close relationship between type of production and presence of virus, as PRV-3 related disease was only observed in rainbow trout produced under RAS conditions, and with 87% PRV-3 positive production farms versus 47% positive broodstock farms, indicating that this virus primarily is a threat to the rapidly growing RAS industry. In Denmark, most broodstock farms are surveyed for the presence of IPN virus and *Renibacterium salmoninarum*, the causative agent of bacterial kidney disease (BKD), in order to produce fish certified free for these pathogens. Therefore, it was a natural step to assess if PRV-3 could be incorporated into this program for producing certified PRV-3-free eyed eggs and progeny for trade. Unfortunately, some of the currently certified BKD and IPN free farms were positive for PRV-3, nevertheless a number of broodstock farms are apparently still free, and might in future form the basis for stocking PRV-3 free production farms, e.g., in fully recirculated aquacultures.

Recently, a role of PRV-3 in the development of PDS in brown trout has been suggested and discussed [17,21]. PDS is a multifactorial disease associated with mass mortality nearing 100% in brown trout in pre-Alpine rivers of southern Germany, Austria, and Switzerland [22], which so far has not been reported in Denmark. Importantly, in our study no signs of disease or mortality were observed in farms producing brown trout for restocking purposes despite findings of PRV-3. As most of brown trout production is aimed for restocking rivers in Denmark, this host may however play a relevant role in maintaining the infection in the river environment, and further studies will be needed to investigate the spread of PRV-3 in wild salmonid populations indigenous in Danish rivers. 

Importantly, no PRV-3 was detected in Atlantic salmon used as broodstock for re-stocking purposes from DCV, these data are in agreement with what has been reported in Norway [28]. PRV-3 is not prevalent in wild populations of Norwegian or Danish Atlantic salmon, and this possibly reflect a lower susceptibility of this host for PRV-3. Due to limited sample size of sea trout tested in this study, no conclusion can be drawn on the prevalence of PRV-3 in this salmonid species in Denmark.

Interestingly, disease outbreaks consistent with the detection of PRV-3 occurred exclusively in rainbow trout in farms with RAS. Fish in RAS with biofilters seem to be at a greater risk of having PRV-3, as 100% of the screened RAS farms with biofilters were positive, compared to 50% of farms with flow through systems. Additionally, disease was observed in 44% of PRV-3 positive RAS farms with biofilter (8 out of 18 farms), while disease has not been observed in flow through farms yet. Previous experimental infection studies [19,20] have shown that PRV-3 replicate in rainbow trout reaching a peak over three to four weeks post exposure, and then clear the infection over time. However, we suggest that common farming practices such as the continuous introduction of new fish batches, and a higher degree of water recirculation enhance the persistence of the infection on the farm site by transferring the virus from one batch to the newly introduced one. The intensity of the introduction of new fish batches as a risk factor is demonstrated by the detection of different PRV-3 isolates from single fish samples in the same farm at the same time point (isolates DK/17-18918 (F), Figure 2a). Furthermore, another farm (shown as (A) in Figure 2a) is represented in the S1 tree by two different isolates collected at different time points. Since the sequencing of the isolates was done with the Sanger method on pooled samples during the surveillance, it is very likely that the co-circulation of different isolates occurs more often than indicated in our results, as we are only showing the most dominant sequence of the viral population at the farm at a given time point. 

Six farms are represented by two to three isolates (Figure 2a) taken at different time points. Three of these farms have identical isolates across time. As an example, one farm was sampled at three different time points (January, August, and November 2018). The isolates from all time points are identical in the S1 segment (marked with (C) in Figure 2a), yet disease was only observed in January. However, the detected viral load was higher in the samples from the two time points with no observed disease compared to the samples taken at the disease outbreak (data not shown). This indicates that the onset of disease happens after the peak of viremia, and thereby, viral detection is not necessarily a suitable predictor of disease. This is in agreement to what is observed during experimental trials, where PRV-3 induced histopathological lesions occurs from two weeks after viral peak [20]. In addition, the sequence identity of PRV-3 isolates detected in the same farm indicate the challenges of removing the pathogen from the RAS farm environment by standard hygiene practices, particularly considering the physical-chemical resistance of naked virus in the environment [29,30]. It should also be considered that multiple factors such as water quality parameters, and presence of other agents might change the clinical picture observed at the farm. 

PRV-3 was first described in 2013 in association with clinical disease outbreak in rainbow trout in Norway [13]. By analysing archived diagnostic samples in our repository, we could trace the presence of PRV-3 in Danish aquaculture back to 1995. Strikingly, sequence analysis show 100% identity in S1 segment with PRV-3 isolates currently detected in the same farm. PRV-1 was first reported in 1999 from a HSMI outbreak in Norway [31]. Later on, archived samples collected in 1977 from steelhead and in 1987 from Atlantic salmon in British Columbia have been shown positive for PRV-1, although the virus was close to the detection limit in the samples from 1977, and was not confirmed by sequencing [32]. This indicates that *Piscine orthoreovirus* has in general been circulating long before being associated with disease. In the case of PRV-3, it is clear that the virus has circulated in Danish farms long before disease onset. One hypothesis to be examined is that the cause of the sudden association with disease could be linked to a more virulent genotype becoming more prevalent. Our data, however, does not support this hypothesis. 

In an attempt to identify virulence markers associated with different PRV-1 strains, Dhamotharan et al. [23] has performed full genome analyses of a panel of 31 sequences representing isolates covering different geographical regions, including Atlantic and Pacific Ocean, temporal distribution (prior to the first description of HSMI), and phenotypic virulence (strains with and without association to HSMI). Phylogenetic analyses of S1 and M2 segment reflect the phenotype of the different viral isolates, supporting that these two segments house virulence marker(s) [23]. As PRV-3 is a sister group to PRV-1, we have tested this hypothesis in the search of disease markers for PRV-3 in Danish aquaculture. Conversely to PRV-1, there is no apparent pattern or grouping of PRV-3 isolates associated with disease in either segment. Isolates that have been associated with disease are identical or very similar to isolates not associated with disease in both segments. In fact, the majority of the Danish isolates are identical or very similar, with a dominant haplotype from which the other isolates differ in only few mutations (Figure 3). Furthermore, sequences of Danish isolates from 1995 are identical to the current surveillance isolates in both segments, indicating that the virus has changed very little in the last 23 years. This could further support that the presence of PRV-3 is necessary but not sufficient to trigger disease, and that other environmental, epidemiological, and host factors are needed to cause a disease outbreak. 

Previous studies have defined two subtypes of PRV-3 based on S1 segment phylogeny, namely PRV-3a which is found in Norway, and PRV-3b which is found in the rest of Europe and Chile [3]. This was however based on just one sequence from a Norwegian PRV-3 isolate. Here, five new sequences from Norwegian isolates from 2016 from rainbow trout were included. These isolates along with the surveillance samples confirm that there are two subtypes of PRV-3, as the same two clades are present in the S1 segment. This is also supported by the analysis of the M2 segment (Figure 2b), as the Norwegian isolates form a separate clade from all other isolates. Four Danish isolates included in this study do not fit the current subtyping particularly for the S1 segment. This includes DK/18-4929-33 and DK/18-6518-30 from rainbow trout, and DK/18-6340-33 and DK/18-5020-3 from brown trout farmed for restocking purposes. Interestingly, DK/18-4929-33 and DK/18-6518-30 are isolates belonging to two farm sites of vertically integrated companies. These sites do not introduce eggs and/or fish from other farms. As for their position in the phylogeny, it is possible to speculate about the origin of these two PRV-3 isolates, and whether this constitutes a separate introduction of PRV-3 in Denmark. 

Analyses of PRV-3 full genomes reveal that L2 is the most variable RNA segment within PRV-3 genome. Furthermore, L2 is the only RNA segment, which shows higher variability in PRV-3 compared to PRV-1. Conversely, segments M2, M3 and S1 (Appendix A) are more variable in PRV-1 compared to PRV-3, but all these results have the caveat that our limited sampling of PRV-1 may not be the best to perform this kind of comparisons. 

The L2 segment encodes for lambda 2 protein, the turret protein protruding around the fivefold axes in the core particles [33]. Experimental reverse genetics on L2 has shown that changes in the turret protein are associated with changes in interferon susceptibility of mammalian reovirus (MRV) [34]. PRV-associated pathology is most likely driven by interferon response in heart tissue, followed by CD8+ lymphocyte infiltration [6,20]. This may indicate that the L2 segment could harbour interesting features associated with virulence in other reoviruses, opening for further examination of PRV. Moreover, being the most variable segment within PRV-3 genome, L2 represent an interesting target for phylogenetic analysis, molecular tracing and study on the evolution of the virus. Further studies should compare the impact of different PRV-3 subtypes in rainbow trout and expand knowledge on pathogen independent factors which enhance disease outbreak. 

## 4. Materials and Methods 

### 4.1. Surveillance

The samples included in this study were obtained from a voluntary targeted surveillance program conducted in 53 Danish aquaculture facilities from late 2017 to early 2019 across 75 samplings. 

The farms included in the program covered a wide variety of aquaculture farms in Denmark, encompassing flow through to fully recirculating aquaculture system. Additionally, farms producing both rainbow trout and/or brown trout were included.

For each sampling, fish displaying clinical signs were selected. If no such fish were present at the time of sampling, healthy fish were selected instead. On average 30 fish were collected at each farm, and sampled individually. Pieces of heart and spleen from each specimen were pooled in tubes with 500 µL RNAlater, and shipped to the laboratory cold. 

Statistical analysis of the test results was performed with Fisher exact test.

### 4.2. Additional Samples

Archived material stored in the freezers of DTU Aqua, Unit for Fish and Shellfish Disease in Lyngby, Denmark was selected for testing for PRV-3. The material consisted of organ homogenate for virological examination collected during clinical disease outbreaks with IPNVin 1995. According to standard operating procedure in the lab, these consisted of a pool of organs (heart, spleen and kidney) from 10 fish homogenized, diluted in MEM 1:10, and stored at −80 °C. 

Additionally, PRV-3 positive samples obtained during a survey in Norway in 2016 [22] were included in our analyses. The samples consisted of heart and spleen from individual fish preserved in RNAlater.

Finally, individual wild fish used as broodstock for re-stocking purposes were obtained from Danish Center for Wild Salmon (DCV) in 2016, 2018 and 2018. Pooled organs (heart, kidney and spleen tissues) from individual Atlantic salmon (*S. salar*) and sea trout (*S. trutta fario*) were screened for PRV-3 by qPCR. Seven sea trout were tested in 2016; 100 Atlantic salmon in 2018, and 30 Atlantic salmon in 2019.

### 4.3. Detection of PRV-3 by RT-qPCR

RNA purification from heart and spleen in RNAlater was done with RNeasy Mini Kit (Qiagen, Hilden, Germany) and QIAcube (Qiagen, Hilden, Germany) according to manufacturer’s recommendations. 

Real-time RT-qPCR for detection of PRV-3 was done as previously described [13], using QuantiTect Probe RT-PCR kit (Qiagen, Hilden, Germany) according to manufacturer’s recommendations in a final volume of 25 µL with 5 µL template. RNase-free water was used as negative PCR control. RT-qPCR was performed using Stratagene Mx3000P and Mx3005P. Data was analyzed in MxPro software (v4.10 build 389, Schema 85, 2007 Stratagene). Cut-off value for positive samples was set at Ct 37. Samples with a Ct > 37 with a sigmoidal shape of the amplification plot were considered suspect. If the amplification plot had a non-sigmoidal shape and Ct > 37, such samples were considered negative. Threshold was set at 0.01 fluorescence (dRn) at a logarithmic y-axis. 

Prior to implementing the diagnostic protocol with 5-fish pools, the sensitivity of testing pooled samples was assessed by pooling samples containing different loads of viral RNA with negative samples (see Appendix A). Tissue lysis was done on individual samples, and the lysate was pooled for RNA extraction. Four types of samples representing strong positive (Ct 18.6-19.8), weak positive (Ct 33.9-35.4), suspect (Ct 37.4-37.5), and negative were included in the testing. Briefly, each sample type was tested after being diluted in the following ratios (v:v): undiluted, 1:1, 1:2, 1:4, 1:6, and 1:9. RNA was purified with RNeasy mini kit (Qiagen, Hilden, Germany) using QIAcube (Qiagen, Hilden, Germany) according to manufacturer’s recommendations. Pooling of material from 5 fish were decided upon for general surveillance and to allow for subsequent “de-pooling”, individual fish tissue lysates were stored on lysis buffer (Qiagen, Hilden, Germany) at –80 °C. If a pooled sample was recovered with very high Ct value or in cases of dubious qPCR results, samples were de-pooled and tested individually. 

For the majority of cases, fish samples belonging to the surveillance program were analyzed in 5-fish pools. Samples obtained from the Norwegian survey and wild salmonids were analyzed as individual fish samples.

### 4.4. Sequencing

Sequencing of S1 was performed using Qiagen OneStep RT-PCR Kit (Qiagen, Hilden, Germany) according to manufacturer’s recommendations with 5 µL of template. The primers used are described in Table 1. Thermal cycler conditions: 50 °C for 30 min, 95 °C for 15 min, (94 °C for 30 s, 70 °C for 30 s (−0.5 °C per cycle), 72 °C for 1 min) for 30 cycles, (94 °C for 30 s, 55 °C for 30 s, 72 °C for 1 min) for 30 cycles, 72 °C for 10 min. Thermal cycler: MiniAmp and MiniAmpPlus (Applied Biosystems, Foster City, CA, USA).

Sequencing of M2 was done similarly to S1 segment, but combining sequencing primers for (1) two partially overlapping amplicons for surveillance samples, and (2) five partially overlapping amplicons for archived samples. The primers used are described in Table 1.

Thermal cycler conditions: 50 °C for 30 min, 95 °C for 15 min, (94 °C for 30 s, 70 °C for 30 s (−0.5 °C per cycle), 72 °C for 1 min and 20 s) for 30 cycles, (94 °C for 30 s, 55 °C for 30 s, 72 °C for 1 min and 20 s) for 30 cycles, 72 °C for 10 min.

PCR products were separated by electrophoresis in 1.2% (S1) and 2% (M2) E-gels (ThermoFisher Scientific, Waltham, MA, USA) using E-gel powerbase version 4 (Invitrogen, Carlsbad, CA, USA) according to manufacturer’s recommendations. DNA was purified with either QIAquick PCR Purification Kit of QIAquick Gel Purification Kit (Qiagen, Hilden, Germany) following manufacturer’s recommendations. Sanger sequencing were performed by Eurofins genomics (Ebersberg, Germany) using the same primers as described in Table 1. 

### 4.5. Phylogenetic Analyses

For phylogenetic reconstruction, the S1 dataset consisted of 79 sequences of 774 nt (including seven PRV-1 and one PRV-2 sequences as outgroup), the M2 dataset consisted of 44 sequences of 1913 nt (including six PRV-1 and one PRV-2 sequences as outgroup). 

Dataset were aligned using Muscle v. 3.8.425 (http://www.drive5.com/muscle; [35]), with standard settings, as implemented in Geneoius Prime v. 2020.1.

The presence of recombination was tested using RDP4 v. 1.00 [25], http://web.cbio.uct.ac.za/~darren/rdp.html), using the RDP [25], GENECONV [36], Chimera [37], MaxChi [38], BootScan [39], SiScan [40], and 3seq [41] methods with standard settings, and indicating that sequences are linear. Recombination events or the lack thereof was corroborated with GARD [26], in the Datamonkey evolutionary server (https://www.datamonkey.org/gard). No evidence of recombination was found within segment S1 with either method. In M2, weak evidence of recombination was found for sequence NOR/060214 with RDP4, but this was not corroborated with GARD.

The best fitting model of substitution, as indicated as the model with the lowest BIC score (Bayesian Information criterion), was estimated in Mega X [42] for each dataset.

Phylogenetic analyses were carried on using the program PhyML v. 3.2.2 (http://www.atgc-montpellier.fr/phyml, [43,44]) as implemented in Geneious Prime v. 2020.1.1 (http://www.geneious.com/), using the best fitting model of substitution obtained in Mega X. For the S1 and M2 segment the K2+G model (Kimura 2 parameters with rate heterogeneity, [45]) was preferred, whereas for M2 segment HKY+G [46,47] was used Four categories were used for the rate heterogeneity, and PhyML freely estimated the gamma model parameters, number of invariable sites, and other parameters of the model. Branch support was estimated by non-parametric bootstrap with 100 repetitions in PhyML. A ML phylogenetic tree was also reconstructed using RaxML as implemented in Geneious Prime v. 2020.1.1 using the GRT+G model and standard settings with 1000 BS replicates. 

Haplotype networks of the S1 and M2 segment were generated using Pop Art [48], using minimum spanning network (ε = 0).

### 4.6. Full Genome Sequencing of Contemporary Samples

Full genome sequence was done in two recent isolates associated with diseases. Isolate DK/18-3659-17 was previously isolated in Nov 2017, and isolate DK/18-3659-15 in March 2018. Both isolates were present in two rainbow trout farms suffering a heavy onset of disease. 

An in vivo challenge trial was conducted to propagate in rainbow trout PRV-3 isolates from disease outbreaks as described in Dhamotharan et al. [3]. Briefly, blood was collected from the caudal vein in heparin tubes, and plasma was obtained at peak of viremia. Total RNA was extracted from 2 mL of pooled plasma originating from two individuals (Ct 25.78 and 26) adding Trizol as previously described [3]. To the total RNA, 0.1 volumes of 3 M sodium acetate (pH 7.5) and 2 × volumes of absolute ethanol were added, and mixed gently. Macrogen (Seoul, Korea) performed library preparation using the TruSeq RNA Library Prep Kit v2 (Illumina Inc., San Diego, CA, USA), followed by whole genome de novo sequencing (101 bases, paired-end reads) on an Illumina HiSeq4000 platform (1/7th lane). Cleaning and removal of Illumina adapters from raw data was done using BBDuk: Adapter/Quality Trimming and Filtering version 38.97., part of the BBTools package (https://sourceforge.net/projects/bbmap/) using the parameters K-mer = 27, ktrim = r, mink = 4, hdis = 1, and quality-trim to Q30 using the Phred algorithm in both left and right sides of the sequence (qtrim = rl). All sequences shorter than 50 nt were discarded. Sequences were de novo assembled using the genome assembler software SPAdes (version 3.10.1). Given the large amount of reads obtained, and the size of the PRV genome, only 5% of the raw data was used for de novo assembly. Contig annotation was done mapping to the reference sequence NOR/060214. Number of reads for each segment is given in Appendix A.

In vivo experimental trial to propagate PRV-3 was conducted following guidelines for welfare of research animals under license number 2013-15-2934-00976. 

### 4.7. Full Genome Sequencing of Archived Samples

Full genome sequencing was carried out for one of the historical samples from 1995. This sample consisted of a pool of organs from rainbow trout homogenized in MEM for virological examination as prescribed by standard operating procedures for virological examination on cell culture (currently given in Commission Implementing Decision 2015/1554/EC). The organ homogenate was stored at −80 °C for 23 years.

In order to propagate the virus, 800 µL of the original sample were mixed with 800 µL anti-IPN (Anti-IPN SP, Immunological and Biochemical Testsystems, Binzwangen, Germany and 40 µL gentamycin (Life technologies, Carlsbad, CA, USA) and stored at 4 °C overnight, and used next day as inoculum for viral propagation in rainbow trout. 

Viral propagation was done using 15 juvenile rainbow trout (50 g) challenged intraperitoneally with 100 µL inoculum each. In vivo experimental trial to propagate PRV-3 was conducted following guidelines for welfare of research animals under license number 2019-15-0201-00159. 

Fish were kept in a 180 L tank at 12 °C temperature. Non-lethal blood samples were collected from the caudal vein after anaesthesia of fish (80 mg/L of benxocaine) and viral load assessed by qPCR. At the peak of viremia, fish were euthanized (overdose of benzocaine 800 mg/L) and bleed. Blood samples without anticoagulant, and organ samples in MEM were collected. Organ samples were immediately stored at −80 °C.

In order to minimise host genetic material, serum was separated from the whole blood, and 280 µL of each serum sample was diluted in the corresponding volume of AVL-Carrier-RNA buffer from the QIAamp Viral RNA Mini kit (Qiagen, Hilden, Germany) before storage at −80 °C. RNA was extracted from serum samples using the QIAamp Viral RNA Mini kit (Qiagen, Hilden, Germany) following manufacturer’s recommendations. RNA was eluted in 60 µL buffer AVE.

DNA removal from the eluted RNA (60 µL) was done using DNA-free DNA Removal Kit (Life Technologies, Carlsbad, CA, USA) according to manufacturer’s recommendations.

A sequencing independent single-primer amplification (SISPA) method was used according to Mohr et al., 2015 with modifications. In brief, first strand cDNA was generated from a 7 µL of the sample using the SuperScript^TM^ IV First-Strand Synthesis System kit (Invitrogen, Carlsbad, CA, USA) using the primers FR20RV-12N (GCCGGAGCTCTGCAGATATCNNNNNNNNNNNN [27]) at a final concentration of 2.5 µM, with the following thermal profile: 10 min at 25 °C, 10 min at 50 °C, 10 min at 85 °C. Synthesis of the second strand cDNA was done using Klenow DNA polymerase (NEB), and final amplification using Phusion High-Fidelity PCR master mix with HF buffer (NEB) with FR20RV primer (GCCGGAGCTCTGCAGATATC) according to previously published protocol [27]. Lastly, PCR products were purified using QIAquick PCR Purification Kit (Qiagen, Hilden, Germany) according to manufacturer’s recommendations.

Libraries were prepared with the Nextera XT DNA Library Preparation Kit (Illumina, San Diego, CA, USA) and sequenced with MiSeq v3 Reagent Kit (300PE) using Illumina MiSeq platform. Sixteen sample libraries were pooled equimolarly, of which only one corresponded to PRV-3. Both library preparation and sequencing were done at DTU core sequencing facilities. 

Cleaning and removal of Nextera adapters from the Illumina MiSeq raw data using BBDuk: Adapter/Quality Trimming and Filtering version 38.97., part of the BBTools package (https://sourceforge.net/projects/bbmap/) using the parameters K-mer = 27, ktrim = r, mink = 4, hdis = 1, and quality-trim to Q30 using the Phred algorithm in both left and right sides of the sequence (qtrim = rl). All sequences shorter than 50 nt were discarded. SISPA primers were removed in both ends of the sequences using the Trim primers function in Geneious Prime v. 2020.1.1 (www.geneious.com), allowing up to 5 mismatches. 

The de novo assembly was performed using Spades v. 3.13.0 (http://cab.spbu.ru/software/spades/). Obtained contigs were mapped to the PRV-3 reference sequence NOR/060214 [3] to identify the different segments. Re-mapping of raw sequences to contigs from the de novo assembling was done using BBMap version 38.37, part of the BBTools package, with, Kmer = 13. Annotations were transferred to the final contig by similitude > 95% to the reference sequence NOR/060214. 

### 4.8. Phylogenetic Analyses of PRV-3 Full Genomes

Phylogenetic analyses were performed including the three full genomes reported in this study, together with the only two full genomes of PRV-3 available in public databases [2,3] (NOR/060214 and ADLPRV3). Additionally, an unpublished, and more or less finished genome from PRV-3 found in the GenBank was also included. Five representatives of PRV-1, spanning a time period from 1988 to 2014 were included, together with the only PRV-2 full genome available so far [11].

Phylogenetic analyses were done for each segment independently, and in a concatenated matrix of all ten segments. As many of the sequences were short in the segments ends, phylogenetic analyses were done including coding regions only. 

Dataset were aligned using Muscle v. 3.8.425 (http://www.drive5.com/muscle; [35]), with standard settings, as implemented in Geneious Prime v. 2020.1.1. The best fitting model of evolution was estimated with Mega X and phylogenetic analyses were done in PhyML v. 3.2.2 as specified in Section 4.6.

Average genetic distance within group (PRV-1 and PRV-3) were calculated with Mega X [42] for each individual segment.

## 5. Conclusions

The surveillance program conducted in Denmark in 2017 to 2019 revealed that there is a high prevalence of PRV-3 in Danish aquaculture. The virus is widespread within the country, and does not follow a natural pattern, as is expected by virus movement related to trade of live fish and eggs. Recirculating aquaculture systems and grow-out facilities seem to be at a greater risk of having PRV-3. Importantly, disease associated to PRV-3 has only been observed in RAS in Denmark, and disease has not been observed in any brown trout tested in the surveillance program despite findings of the virus. Additionally, PRV-3 has not been detected in wild populations of brown tout or Atlantic salmon.

In order to backtrack PRV-3 in Denmark, samples from diagnostic cases from 1995 were tested for PRV-3, showing that PRV-3 has been present in Danish aquaculture since at least 1995.

Phylogenetic analyses of segments S1 and M2 supports that there are two subtypes of PRV-3, with PRV-3a and PRV-3b being found in Norway and in Europe and Chile, respectively. Importantly, these two segments do not reveal any direct association between sequence and disease. Sequences from samples from 1995 are also identical to the contemporary sequence isolates, indicating that the virus has not changed much in these two segments for the past 23 years, further supporting that external factors are necessary for the development of disease. Importantly, the virus has been circulating long before the association with disease, and while it could be that the cause of sudden association with disease may be linked to a more virulent genotype becoming prevalent, our data does not support such a hypothesis. 

## Figures and Tables

**Figure 1 pathogens-09-00823-f001:**
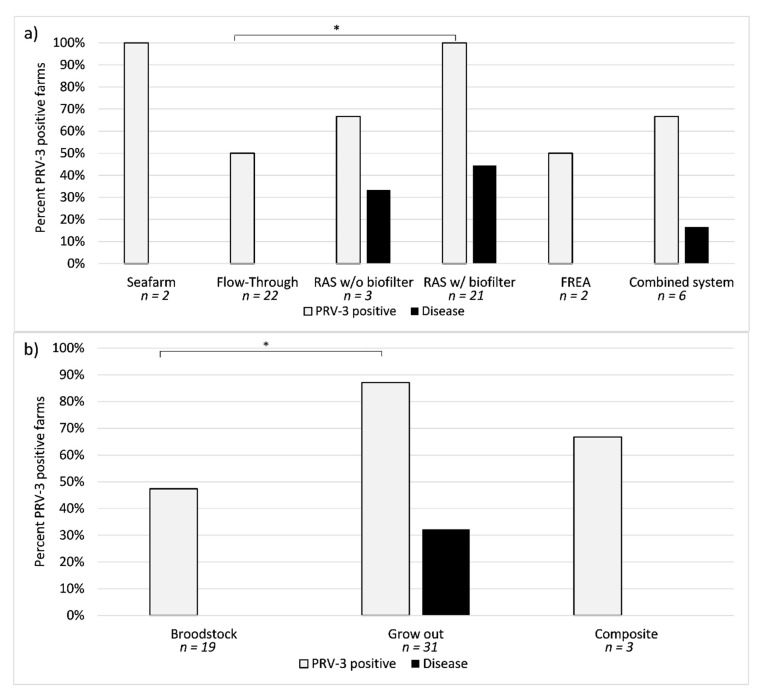
On the y-axis percent PRV-3 positive farms; on the x-axis the categories investigated and the number of farms tested. If a farm has been tested multiple times with one test being positive for PRV-3, it is recorded as positive here. (**a**) Distribution of PRV-3 positive farms in relation to water recirculation intensity, and prevalence of disease associated with PRV-3. RAS w/o biofilter = RAS without biofilter (high exchange of water), RAS w/ biofilter = RAS with biofilter (low water exchange), FREA = Fully recirculated indoor aquaculture system, combined systems = farms that have both flow through and RAS. Prevalence of PRV-3 in flow through estimated to 0.500 (95% CI; 0.3072–0.6928), and in RAS w/ biofilters 1.000 (95% CI; 0.8241–1.000). (**b**) Distribution of PRV-3 positive farms according to the farm production type, and prevalence of disease associated with PRV-3. Prevalence of PRV-3 in broodstock farms estimated to 0.4737 (95% CI; 0.2733–0.6829), and in grow-out 0.8710 (95% CI; 0.7115–0.9487). * = *p* < 0.01.

**Figure 2 pathogens-09-00823-f002:**
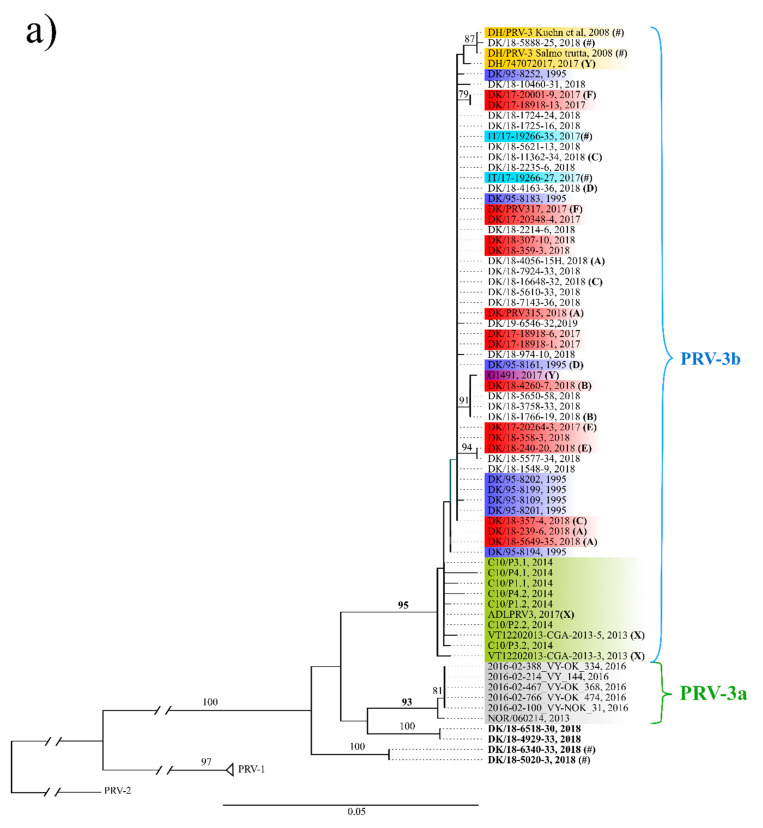
Maximum likelihood tree (PhyML) of the (**a**) S1 and (**b**) M2 segment of PRV-3. Isolates are marked as following: Red, Danish isolates associated with disease; blue, Danish isolates from 1995; yellow, German isolates; green, Chilean isolates; light blue, Italian isolates; grey, Norwegian isolates; purple, Scottish isolates; #, brown trout; X, coho salmon; Y, Atlantic salmon; **bold**, sequence isolates that change position. Isolates originating from the same farm sampled at the same or different time points are marked with (A), (B), (C), (D), (E), and (F). Subtyping of PRV-3a and -3b is based on the S1 segment only. The PRV-3a clade consists of Norwegian isolates only, while the PRV-3b clade consists of isolates from Europe and Chile.

**Figure 3 pathogens-09-00823-f003:**
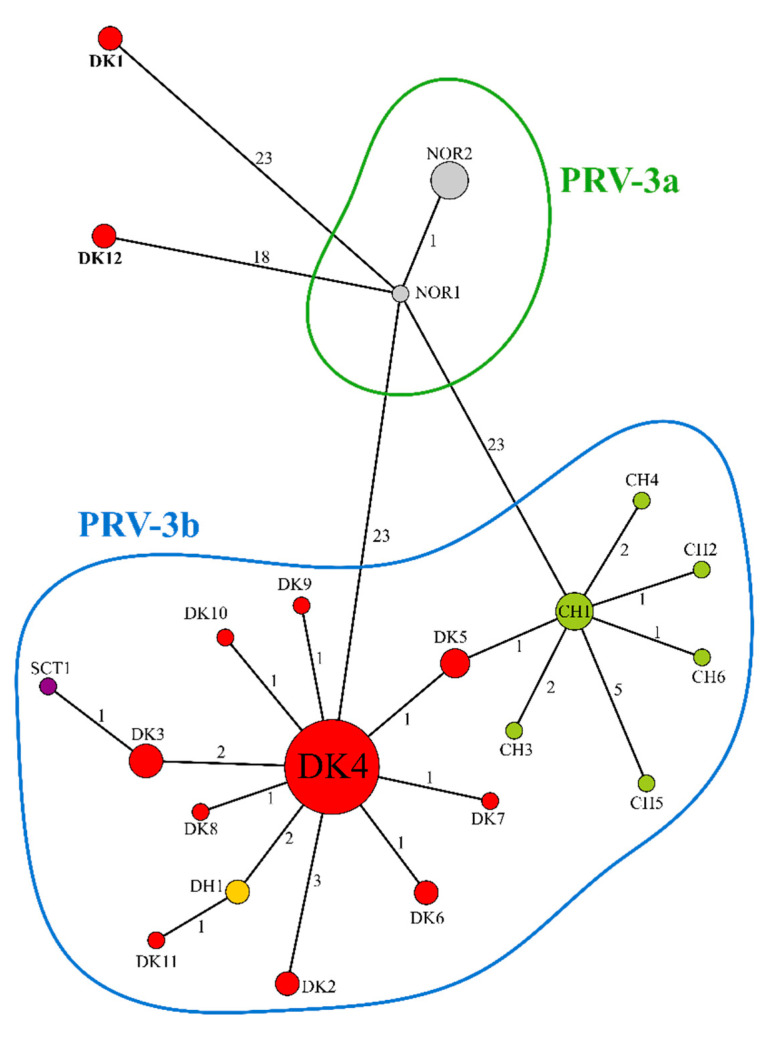
Minimum spanning network of the S1 segment. Red, Danish haplotypes (DK); green, Chilean haplotypes (CH); grey, Norwegian haplotypes (NOR); yellow, German haplotypes (DH); purple, Scottish haplotypes (SCT). See Appendix A for isolates, species, geographic distribution, and time span of each haplotype.

**Figure 4 pathogens-09-00823-f004:**
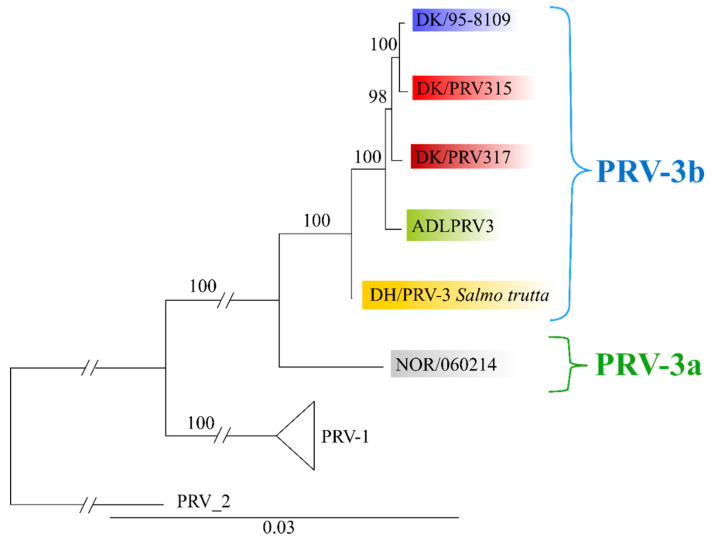
Full genome concatenated phylogenetic tree of *Piscine orthoreovirus* (PRV) genomes. Collapsed node consists of five PRV-1 isolates.

**Table 1 pathogens-09-00823-t001:** Primers for S1 and M2 sequencing.

Sample Type	Segment	Primer Name	Sequence
Surveillance and archived	S1	S1ORF_F [13]	ATGGCGAACCATAGGACGGCGA
		S1ORF_R [13]	TCACGCCGATGACCACTTGAGCA
Surveillance	M2	M2-1 [17]	ATTTTGGGTAACTGGCGACG
		M2-1084R	TGTGATTGCCATGCTGGTGA
	M2	M2-930F	CTGTGACAACACCCTGGGTT
		M2-1R [17]	GTACAACGTACTGCCAGGGT
Archived	M2	M2-1F	CATTTGTTTAACAGGCTTGACC
		M2-453R	CGCAGGTTGATCACACTTGC
	M2	M2-434F	GCAAGTGTGATCAACCTGCG
		M2-949R	AACCCAGGGTGTTGTCACAG
	M2	M2-930F	CTGTGACAACACCCTGGGTT
		M2-1545R	AGCAGTCGTGTAATTGCCGA
	M2	M2-1505F	AGAGTCATTCCTGCTGGTGC
		M2-2059R	CCAGACGCCTAGCTTCCTTT
	M2	M2-1505F	AGAGTCATTCCTGCTGGTGC
		M2-1776R	GAGGCAGCCTTGTCTGACAT

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
