# Peer review of "Emergence and Spread of Piscine orthoreovirus Genotype 3"

_pathogens, 2020, doi:10.3390/pathogens9100823_

Round 1
Reviewer 1 Report
The MS entitled Emergence and Spread of Piscine orthoreovirus
3 Subtype 3" describes a molecular detection work of the virus in some Danish salmonid farms, in order to improve a better understanding of the epidemiology of PRV-3 in Denmark. I believe this is a very useful work enhancing our knoweldge on the distribution of this virus in some regions including European aquaculture sector especially in biofiler and RAS systems.
The data also help to a better undestanding of source of the virus, although sitll more works are required to know about the natural origin source of virus.
Paper is well organised and with a comperenshive discussion.
Author Response
Response to reviewer 1 comments
Points:
The MS entitled Emergence and Spread of Piscine orthoreovirus Subtype 3" describes a molecular detection work of the virus in some Danish salmonid farms, in order to improve a better understanding of the epidemiology of PRV-3 in Denmark. I believe this is a very useful work enhancing our knowledge on the distribution of this virus in some regions including European aquaculture sector especially in biofiler and RAS systems.
The data also help to a better understanding of source of the virus, although still more works are required to know about the natural origin source of virus.
Paper is well organized and with a comprehensive discussion.
Authors Response (AR): we appreciate the comment of reviewer 1.
Point: English language and style are fine/minor spell check required
Authors Response (AR): language revision has been done, and minor spell errors corrected
Reviewer 2 Report
The paper Pathogens_ 863229 takes into consideration an emerging viral agent for Salmonids from several points of view.
The manuscript appears particularly complex in its final draft and is not always readily usable for reading. I do not see major remarks to be attributed to the authors, except that of having indicated the bibliographical citations in a very coarse way; it is a serious flaw that must be completely remedied by following the indications of the journal.
For these reasons, in my opinion, the paper needs a profound revision of the bibliography and the authors must try to make the text more usable as a whole.
Author Response
Reviewer 2
The paper Pathogens_ 863229 takes into consideration an emerging viral agent for Salmonids from several points of view.
The manuscript appears particularly complex in its final draft and is not always readily usable for reading. I do not see major remarks to be attributed to the authors, except that of having indicated the bibliographical citations in a very coarse way; it is a serious flaw that must be completely remedied by following the indications of the journal.
For these reasons, in my opinion, the paper needs a profound revision of the bibliography and the authors must try to make the text more usable as a whole.
Authors Response (AR): we appreciate the comment of reviewer 2. We have gone through the manuscript once again and update the reference list. To the best of our knowledge, the way references are included as well as the reference list, reflects the guidelines of the journal. We are at full disposal for further amendments if needed.
Reviewer 3 Report
The manuscript entitled ‘Emergence and Spread of Piscine orthoreovirus Subtype 3’ used molecular surveillance and sequencing to document the epidemiology of PRV-3 in farmed salmon as well as wild fish in Denmark.
The discovery of PRV-3 is relatively recent, so this study is an important addition to the literature. Overall, the study is well designed, and will be an important contribution. In my review below, I highlight some major issues which need to be resolved before publication, and I hope my criticism is constructive.
Of course, I have inevitably focussed on the negatives, but overall, this is a good contribution, and I look forward to seeing it published.
Firstly, and very importantly, is the issue of data availability. There are no accession numbers listed for sequences from this study (see https://www.mdpi.com/journal/pathogens/instructions#sequence). The sequences need to be made publicly available before this is published. I assume that the data would be released in the final version, and table S2 will be updated with all the accession numbers. I emailed the editor asking for sequences but only received them the day of the review deadline. All sequence data needs to be publicly available before publication. As a reviewer, I wanted to look at the sequences a bit more closely, to see if I could check some of the potential issues that I commented on below (e.g. recombination).The full genome sequences and sequencing reads should be uploaded to genbank and the sequence read archive respectively.
The description of different levels of variation in PRV can be confusing to the reader. I recommend that the authors define terms early on in the manuscript, and stick to them. I recommend using strain to describe PRV-1, 2 and 3 and sub-strain to describe variation within these groups (e.g. sub-strain PRV-3a). Try to avoid the term issolate to describe a sequence. ‘Sequence isolate’ or ‘sequence variant’ might be better, as you did not actually isolate virus so this might be confusing to some.
I am afraid I have several issues with figure 1, which needs improvement. Firstly, and this is getting the basics right, the y-axis is not labeled. As the figure is currently, it is hard to comprehend the results. Plotting the number of samples within each category does not communicate the results well. I find the choice of plotting the number of samples tested rather than the actual proportion positive an odd choice, and not fit for publication. It would be much clearer to plot the proportion (i.e. percentage) positive, and label the bar with the number of samples (n=xx). Binomial proportion confidence interval (e.g. Wilson’s) can be used to show the confidence intervals. A second plot, or different colour bar can be used to show the proportion with disease in the same way.
There were also some issues with the phylogeny. Bootstraps (BS) should be placed on the node, not on the branch, as this makes it hard to tell which branching event they are referring to.
You state that PRV-3a and PRV-3b diverged with a BS of 93% in Figure 2A. By looking at your tree, the divergence between these two sub-strains is earlier in the tree, at a node with no BS shown.
In Figure 2b, what made you decide to include sample NOR-060214,2013 as part of PRV-3a, the group is not monophyletic. Line 177 - 84% Bootstrap - not sure where you are getting this from? There is no 84% value in the tree.
In line 166 you state a 83% bootstrap, there is no value with this number on the tree.
Line 179: what do you mean by ‘poor phylogenetic signal’. More detail needed here. Do you mean each segment shows a different phylogeny?
Did you check for recombination (using RDP4 or similar) to investigate the cause of the sequences that lie inbetween the two sub-strains. A very quick look at the sequences shows that this clade shares some genomic features with both 3a and 3b, so could potentially be a result of recombination, or just divergence. I think it would be worth checking for recombination using software, and then at least you might be able to offer a better explanation than just ‘poor phylogenetic signal’. In your supplementary tree you include these sequences in 3b - all figures need to be consistent with the labelling of clades.What do the asterisks on the tree mean?
Discussion comes to an abrupt end. A summary/ conclusions paragraph might improve things.
A map of where all the samples are from could be useful.
The conflict of interest statement should mention that one of the authors works for the industry (Danish Aquaculture Organisation (Dansk Akvakultur)). I do not think there is any foul play here, but it is important to mention any potential conflicts.
Other comments by line number:
Line 80 - not strictly true - PRV-1a has been associated with HSMI (in British Columbia). Better to discuss potential differences in virulence, whereas this statement implies differences in pathogenicity. These terms are not interchangeable.
Ln 101 more details on suspicion of disease
Ln 103. How many wild fish were screened, this needs to be more clear in the paper, or maybe I missed it.
Ln 206 - grammar needs revising - more clear to say that it ‘indicates a single introduction event’?
Ln 209 - useful to give the percent nucleotide identity here.
Ln 228. change assembling to assembly
- Change to The de novo assembly
- Grammar need addressing
- Change ‘was’ to ‘were’
- Change ‘natural’ to ‘obvious’, or a a ‘natural step’
- Change to ‘what has been reported in Norway’
- Change to ‘populations’
- I agree with your point, but not sure consensus is strictly the right word here. More likely you sequenced the most dominant sequence.
- The use of the word similarly here suggests that the archived samples in BC also had 100% sequence identity - I don’t think this is what you mean.
- The positive detection in 1977 is putative, not verified by sequencing, and near the limit of detection.
- I agree, but older than who first thought? - needs citation.
- PRV-1a in the Pacific has been associated with disease, so this needs correcting here. Virulence is not the same as pathogenicity.
- This analysis is not included in the paper. I think a supplementary figure is needed showing the variability calculations. What is the limited sampling of PRV-1 you refer to? The sequences you selected, or sequences available for PRV-1.
- Grammar confusing. Why did you only use 5% of the data for assembly? Or do you mean the assembled contigs were made up of 5% of total reads?
- Change to ‘carried out’
- De Novo?
Reviewer 4 Report
I find surveys like this one very helpful in defining the distribution of viruses. I offer comments and suggestions to improve the manuscript (I also include some copy-editing comments, but these are not comprehensive):
Line 41 – I do not recommend citing reference 5 because the association of PRV with Jaundice in British Columbia has not been confirmed under controlled laboratory conditions; however, if reference 5 is retained, the following papers should also be cited:
Garver, K. A., Marty, G. D., Cockburn, S. N., Richard, J., Hawley, L. M., Muller, A., . . . Saksida, S. (2016). Piscine reovirus, but not jaundice syndrome, was transmissible to Chinook Salmon, Oncorhynchus tshawytscha (Walbaum), sockeye salmon, Oncorhynchus nerka (Walbaum), and Atlantic Salmon, Salmo salar L. Journal of Fish Diseases, 39, 117-128. doi: 10.1111/jfd.12329
Purcell, M. K., Powers, R. L., Taksdal, T., McKenney, D., Conway, C. M., Bilott, D. G., . . . Winton, J. (2020). Consequences of Piscine orthoreovirus genotype 1 (PRV-1) infections in Chinook salmon (Oncorhynchus tshawytscha), coho salmon (O. kisutch) and rainbow trout (O. mykiss). J. Fish Dis., 43, 719-728. doi: 10.1111/jfd.13182
Lines 41 and 49 (and elsewhere) – replace “associated to” with “associated with” [as in line 48]
Line 55 – I am not convinced that a cause-and-effect relation between PRV-3 and disease has been clearly demonstrated. I recommend revising the text accordingly. For example, I am not able to find in reference 17 Table 3 any mention of whether the negative control fish had heart lesions; also, in Table 3 of reference 17, the very small numbers of fish with lesions with the PRV-3 purified particles are not enough to be statistically different from 0. Lesions are most common in the 10-week group cohabitated with fish injected with infected blood; if these fish were all from one tank, the study might have detected a tank effect rather than a PRV-3 effect. The references cited seem to best support the conclusion that PRV-3 infection is independent of disease (i.e. a relatively common virus that sometimes occurs in fish that have disease caused by something else).
Line 103 – replace “wild salmonid” with “wild salmonids”
Line 110 – if only five farms tested negative across multiple samplings, does that mean that 10 farms were tested only once? What was the distribution of positive samples (i.e, how many were positives on single or multiple samples)?
Figure 1 – for clarification, what is the y-axis? Is it the number of farms tested for each category? Or, the number of samples groups tested? Or, something else? For example, if a farm was tested three times and was positive one time and negative two times, would results from that farm appear only in the positive bar or both bars? Also, is “disease” status here any disease (e.g., the farm had increased mortality and then PRV-3 testing occurred), or were other causes of disease ruled out and PRV-3 is suspected to be the cause of the disease, or something else?
Lines 141 and 142 – The first sentence repeats the methods and may be deleted.
Lines 255 - 258 – If Pathogens style is to lead the Discussion section with the study objectives, then this is acceptable. However, my preference is to list the objectives of the study in the Introduction section and then summarize how those objectives have been met in the Discussion section (as is done here beginning at the end of line 258.
Line 273 – Has the hypothesis that PRV-3 is derived from the source hatchery/hatcheries been considered? For example, if all farms using RAS systems source their fish from PRV-3 positive hatcheries, and farms using other systems source their fish from PRV-3 negative and positive hatcheries, then the differences in occurrence of PRV-3 might be a result of source hatchery rather than production system.
Lines 284 – 285 – Can this sentence be revised to focus on Discussion rather than repeating methods?
Line 324 – I recommend replacing “pathology occurs” with “lesions occur”
Line 348 – replace “to what observed for PRV-1,” with “to PRV-1”
Line 354 – 356 – good point
Line 502 – replace “carried on” with “carried out”
Line 517 – Instead of “ad bleed”, is “and bled” what is meant?
Author Response
Reviewer 3.
I find surveys like this one very helpful in defining the distribution of viruses. I offer comments and suggestions to improve the manuscript (I also include some copy-editing comments, but these are not comprehensive):
Point 1. Line 41 – I do not recommend citing reference 5 because the association of PRV with Jaundice in British Columbia has not been confirmed under controlled laboratory conditions; however, if reference 5 is retained, the following papers should also be cited:
Garver, K. A., Marty, G. D., Cockburn, S. N., Richard, J., Hawley, L. M., Muller, A., . . . Saksida, S. (2016). Piscine reovirus, but not jaundice syndrome, was transmissible to Chinook Salmon, Oncorhynchus tshawytscha (Walbaum), sockeye salmon, Oncorhynchus nerka (Walbaum), and Atlantic Salmon, Salmo salar L. Journal of Fish Diseases, 39, 117-128. doi: 10.1111/jfd.12329
Purcell, M. K., Powers, R. L., Taksdal, T., McKenney, D., Conway, C. M., Bilott, D. G., . . . Winton, J. (2020). Consequences of Piscine orthoreovirus genotype 1 (PRV-1) infections in Chinook salmon (Oncorhynchus tshawytscha), coho salmon (O. kisutch) and rainbow trout (O. mykiss). J. Fish Dis., 43, 719-728. doi: 10.1111/jfd.13182
Authors Response (AR): we appreciate the reviewer comments and have updated the reference list as well as the text. It now reads.
“ To date, three subtypes of PRV have been characterized: PRV-1 has shown to cause heart and skeletal muscle inflammation (HSMI) in farmed Atlantic salmon in Norway [4]; and has been detected in melanised foci in the fillet of farmed Atlantic salmon in Norway [5].
Furthermore, PRV-1 has also been associated to jaundice syndrome in chinook salmon (O. tshawytscha) [6] , but this has not been confirmed by experimental studies [7,8].”
Pont 2 Lines 41 and 49 (and elsewhere) – replace “associated to” with “associated with” [as in line 48]
Authors Response (AR): agree, this has changed throughout the text.
Point 3. Line 55 – I am not convinced that a cause-and-effect relation between PRV-3 and disease has been clearly demonstrated. I recommend revising the text accordingly. For example, I am not able to find in reference 17 Table 3 any mention of whether the negative control fish had heart lesions; also, in Table 3 of reference 17, the very small numbers of fish with lesions with the PRV-3 purified particles are not enough to be statistically different from 0. Lesions are most common in the 10-week group cohabitated with fish injected with infected blood; if these fish were all from one tank, the study might have detected a tank effect rather than a PRV-3 effect. The references cited seem to best support the conclusion that PRV-3 infection is independent of disease (i.e. a relatively common virus that sometimes occurs in fish that have disease caused by something else).
Authors Response (AR) to the editor: We have to underline that the current manuscript aims at investigating prevalence and genetic features of PRV-3 in salmonids. The text now reads “ A first experimental work has shown efficient horizontal transmission of PRV-3 and development of heart pathology resembling HSMI in rainbow trout after viral peak in host target organs [18]. The same study showed a certain host preference for rainbow trout compared to Atlantic salmon. Furthermore, development of heart pathology resembling HSMI was observed after viral peak in target organs in an experimental infection with purified viral particles in rainbow trout [19] further corroborating the role of PRV-3 as causative agent of heart pathology in Rainbow trout.”
Authors Response (AR) to the reviewer comments.
In relation to the comment raised from the reviewer we would like to underline that there are, currently, two published and independent experimental studies investigating the pathogenesis of PRV-3 in Rainbow trout. PRV-3 replicates efficiently in rainbow trout erythrocytes and therefore the virus RNA is detectable in highly perfused organs such as heart and spleen, once the virus reach the peak of it replication, IFN response is triggered, subsequently heart pathology is observed in correlation with hyper-expression of lymphocytic cytotoxic markers. It is important to underline that in both studies, heart pathology occurs after the viral peak. Factors affecting severity of pathology as well as number of fish showing heart pathology per time point can be related to presence of virulence markers, individual host factor related to inflammatory response, dose provided in the experimental infection (particularly low in the study with purified particles) etc. It has to be noted in the study with purified viral particles NO HSMI like signs were ever observed in control fish that, while sparse and mild findings in the heart NOT consistent with HSMI, affecting the interface between compactum and spongiosum layers, were observed across the experimental groups.
Although we fully acknowledge that the severity of the pathology and development of disease can be also affected by a number of factors which are not directly linked to the pathogen, the timing in the development of pathology as well as immune response activation does not support the conclusion suggested by the reviewer “(i.e. a relatively common virus that sometimes occurs in fish that have disease caused by something else)” but, on the contrary, comply with the hypothesis that PRV-3 is responsible for development of heart pathology in rainbow trout. .
Point 4. Line 103 – replace “wild salmonid” with “wild salmonids”
Authors Response (AR): agree (done)
Point 5. Line 110 – if only five farms tested negative across multiple samplings, does that mean that 10 farms were tested only once? What was the distribution of positive samples (i.e, how many were positives on single or multiple samples)?
Authors Response (AR): Yes 10 farms tested negative at one sampling. 5 farms have been tested multiple times and remained negative. It has occurred 4 times that farm tested negative and then positive.
Point 6. Figure 1 – for clarification, what is the y-axis? Is it the number of farms tested for each category? Or, the number of samples groups tested? Or, something else? For example, if a farm was tested three times and was positive one time and negative two times, would results from that farm appear only in the positive bar or both bars? Also, is “disease” status here any disease (e.g., the farm had increased mortality and then PRV-3 testing occurred), or were other causes of disease ruled out and PRV-3 is suspected to be the cause of the disease, or something else?
Authors Response (AR): Caption of the figure amended, now it reads Figure 1: On the “y” axis number of farms; on the “x” axis the categories investigated and the number of farms tested. If a farm has been tested multiple times and one of these testing is positive for PRV-3 the farm is counted only in the positive bar of the histogram. a) Distribution of PRV-2 3 positive farms in relation to water recirculation intensity, and prevalence of disease associated to with PRV-3. RAS w/o biofilter = RAS without biofilter (high exchange of water), RAS w/ biofilter = RAS with biofilter (low water exchange), FREA = Fully recirculated indoor aquaculture system, combined systems = farms that have both flow through and RAS. b) Distribution of PRV-3 positive farms according to the farm production type, and prevalence of disease associated to with PRV-3.
Point 7. Lines 141 and 142 – The first sentence repeats the methods and may be deleted.
Authors Response (AR): we prefer to leave the sentence. In Pathogens, results sections comes before M&M therefore we prefer to introduce the resulst with a brief sentence on the methods applied to obtain such results
Point 8. Lines 255 - 258 – If Pathogens style is to lead the Discussion section with the study objectives, then this is acceptable. However, my preference is to list the objectives of the study in the Introduction section and then summarize how those objectives have been met in the Discussion section (as is done here beginning at the end of line 258.
Authors Response (AR): we don’t have further comments on this
Point 9. Line 273 – Has the hypothesis that PRV-3 is derived from the source hatchery/hatcheries been considered? For example, if all farms using RAS systems source their fish from PRV-3 positive hatcheries, and farms using other systems source their fish from PRV-3 negative and positive hatcheries, then the differences in occurrence of PRV-3 might be a result of source hatchery rather than production system.
Authors Response (AR): Fish is meant broadly as live fish or non disinfected eyed eggs (now updated in the text)
Point 10. Lines 284 – 285 – Can this sentence be revised to focus on Discussion rather than repeating methods?
Authors Response (AR): amended
Point 11. Line 324 – I recommend replacing “pathology occurs” with “lesions occur”
Authors Response (AR): amended
Point 12. Line 348 – replace “to what observed for PRV-1,” with “to PRV-1”
Authors Response (AR): amended
Point 13. Line 354 – 356 – good point
Authors Response (AR): No comment
Point 14. Line 502 – replace “carried on” with “carried out”
Authors Response (AR): amended
Point 15. Line 517 – Instead of “ad bleed”, is “and bled” what is meant?
Authors Response (AR): amended as requested. Blood samples were collected from the fish after euthanasia.
Round 2
Reviewer 3 Report
Thank you to the authors for their clear and concise response. I am satisfied with the improvements made to the manuscript and all of my major concerns have been addressed. A few minor points remain to be addressed (see attached .doc.
